# Short-Term Responses of Alpine Vegetation to the Removal of Dominant versus Sparse Species

**DOI:** 10.3390/plants13131756

**Published:** 2024-06-25

**Authors:** Weichao Wang, Wei Li

**Affiliations:** Soil and Water Conservation Institute, Southwest Forestry University, Kunming 650224, China; wangweichao1224@163.com

**Keywords:** alpine grassland, dominant removal, community impacts, ecosystem impacts, mass ratio hypothesis

## Abstract

The mass ratio hypothesis posits that ecosystem functions are predominantly influenced by the dominant species. However, it remains unclear whether a species must be abundant to exert functional dominance. We conducted a removal experiment in an alpine grassland near Pudacuo National Park, Yunnan, China, to assess the community and ecosystem impacts of the removed species. We implemented four treatments as follows: exclusive removal of the most abundant species (*Blysmus sinocompressus*), exclusive removal of a sparse species with high individual biomass (*Primula secundiflora*), simultaneous removal of both species, and a control with no removals. Results showed that removing *B. sinocompressus* significantly reduced biomass production, supporting the mass ratio hypothesis, while removal of *P. secundiflora* had negligible effects. *B. sinocompressus* removal positively impacted community metrics like coverage, species evenness, and the Shannon diversity index, but not species richness, likely due to its spatial dominance. Conversely, *P. secundiflora* removal had minimal community impact, probably due to its limited influence on nearby species. This study underscores the proportionate roles of the dominant species in alpine grasslands, emphasizing that their community and ecosystem impacts are proportional to their abundance.

## 1. Introduction

Ecological communities commonly display dominance by a few highly numerous species, often referred to as the dominant species. According to the mass ratio hypothesis [1], the dominant species exhibit large effects on community structure and ecosystem functioning. For example, past studies have consistently demonstrated that the dominant species can exert a strong influence over the structural characteristics of a community [2,3,4]. Often, they outcompete other species by occupying and utilizing a majority of the resources within that community [5,6,7]. Alternatively, the dominant species may modify stressful environmental conditions and facilitate the survival of other species [8,9,10]. Similarly, root structure and carbon allocation capacity of the dominant species may affect soil carbon storage and nitrogen cycling, with important implications for productivity and other ecosystem functions [11,12]. While the dominant species play a leading role in shaping ecosystem functions due to their predominant contribution to biomass, and it is commonly assumed that functionally dominant species also exhibit high abundance, certain non-dominant species may also play important ecological roles disproportional to their abundance [4,13]. Therefore, it remains unclear whether abundance dominance is a prerequisite for a species to be functionally dominant. For example, it is unclear whether certain sparse species, which are limited in abundance but with high individual biomass, can exert disproportionately large community and ecosystem impacts.

An effective method for assessing the impact of a species on its surrounding environment, community, and ecosystem is by conducting removal experiments. Removal experiments are generally performed to assess population and community responses after removal [6]. These experiments entail the removal of certain species (often the dominant species), followed by the observation of how the remaining species respond. This approach helps researchers infer the relationships between the removed species and the remaining species of that community. If the remaining species increase after the removal treatment, it suggests that the removed species may exert a strong inhibitory effect on other species. Moreover, while manipulating species compositions and abundances in artificially assembled ecosystems can help us understand the ecological consequences of species and diversity losses, the assembly process of these communities is inherently characterized by randomness, and their relevance to real ecosystems continues to be intensely debated [14,15]. By contrast, removal experiments are often conducted in naturally assembled communities characterized by non-random species composition and relative abundance, rendering them more valuable for comprehending the community and ecosystem impacts of the removed species [9,16,17].

The alpine grassland ecosystem is characterized by harsh environmental conditions and unique species composition [18]. Within this system, the dominant species often play crucial roles in maintaining ecosystem functions. For instance, the removal of the dominant species can lead to substantial changes in community composition and ecosystem processes [19,20]. In contrast, the role of sparse species, which have high individual biomass but low abundance, is less understood, although these species may contribute to ecosystem functioning in ways that are not directly proportional to their abundance [21,22]. To address this gap, we conducted a removal experiment in the natural alpine grasslands of Shangri-La, which is situated in the southeastern part of the Tibetan Plateau. The primary objective of the present study is to compare the short-term community and ecosystem responses to the removal of a dominant species, *Blysmus sinocompressus* Tang & F.T.Wang, and a sparse species, *Primula secundiflora* Franch, which is with high individual biomass but has low abundance. Specifically, we want to address the following questions: (1) Does the removal of the abundant species and/or the sparse species have distinct community and ecosystem impacts? (2) Could short-term biomass compensation be observed following removal treatments, and are there differences in biomass compensation among the removal treatments? By elucidating the functional roles of the dominant versus sparse species in alpine grassland ecosystems [22], we seek to advance relevant ecological theories and inform conservation strategies in such alpine environments.

## 2. Results

More biomass was removed from the AR and DR treatments, in comparison to the SR treatment (F_2,15_ = 16.31, *p* < 0.01). The removal treatments generally imposed significant community impacts over the alpine grasslands. There was no significant difference in vegetation coverage before the removal treatments (F_3,20_ = 0.57, *p* = 0.59). However, the vegetation coverage significantly decreased months later, following the removal treatments (F_3,20_ = 92.02, *p* < 0.01), with the lowest and highest coverage found in the DR and NR treatment areas, respectively (Figure 1). Also, the vegetation coverage of the AR treatment was significantly lower than that of the SR treatment (Figure 1). After adjustment for the effects of the removed biomass, there was a statistically significant difference in the vegetation coverage between the treatment groups, and the interaction effect between the removed biomass and the removal treatment was not significant (F_2,12_ = 33.45, *p* < 0.01, Table 1).

There was no significant difference in Pielou’s evenness before the removal treatments (F_3,20_ = 0.57, *p* = 0.64), but Pielou’s evenness was relatively high for the AR and DR treatments and remained low for the SR and NR treatments (F_3,20_ = 10.44, *p* < 0.01; Figure 2). After adjustment for the effects of the removed biomass, there was a statistically significant difference in Pielou’s evenness between the treatment groups, and the interaction effect between the amount of biomass removed and the removal treatments was not significant (F_2,12_ = 6.90, *p* = 0.01, Table 2).

Similarly, there was no significant difference in the Shannon diversity index before removal treatments (F_3,20_ = 0.34, *p* = 0.80), but the Shannon diversity was relatively high for the AR and DR treatments but remained low for the SR and NR treatments (F_3,20_ = 23.08, *p* < 0.01; Figure 3). After adjustment for the effects of the removed biomass, there was a statistically significant difference in the Shannon diversity between the treatment groups, and the interaction effect between the amount of biomass removed and the removal treatments was not significant (F_2,12_ = 4.20, *p* = 0.03, Table 3). However, there was no significant difference in species richness either before (F_3,20_ = 23.08, *p* = 0.89) or after (F_3,20_ = 1.71, *p* = 0.21) the removal treatments.

The removal treatments also significantly affected aboveground biomass of the alpine grasslands, with the highest biomass found in the NR treatment, while the lowest biomass was found in the AR and DR treatments (F_3,20_ = 85.29, *p* < 0.01; Figure 4). After adjustment for the effects of the amount of removed biomass, there was a statistically significant difference in the final biomass between the treatment groups (F_2,12_ = 12.55, *p* < 0.01, Table 4). By contrast, there was no significant difference in soil N and P content among the treatment groups (for N: F_1,16_ = 25.45, *p* = 0.74; for P: F_1,16_ = 0.03, *p* = 0.96, Table 5 and Table 6).

## 3. Discussion

Previous studies have suggested that the removal of the dominant species can help enhance plant diversity [4,23]. Our experimental findings partly support this notion. As the removal experiment progresses, the reduction in species dominance means a significant alleviation of competitive pressure for the remaining species. This, in turn, fosters an increase in species evenness (e.g., Pielou’s evenness index) and overall diversity (e.g., Simpson’s diversity index). Also, our results showed that higher values of Pielou’s evenness and Simpson’s diversity index were observed in the AR/DR treatments compared to SR treatment. Given the spatial dominance of *Blysmus sinocompressus*, it is likely that its removal would open up opportunities for other species. Specifically, some native species, such as *Oenanthe javanica* DC. and *Pedicularis sylvatica* L., which were initially absent from the experimental sites, might have taken advantage of the niche vacancies created by the disappearance of *B. sinocompressus* to acquire more nutrients [24,25]. This could have facilitated their seed germination and promoted vegetation growth. Consequently, they successfully established themselves in the experimental sites and gradually expanded their population ranges. By contrast, the removal of *Primula secundiflora* did not yield comparable community impacts, as the dominance structure of the community by *B. sinocompressus* remained unaltered. However, no significant effects of the removal treatments on species richness of the alpine communities were observed. One explanation is that species richness may not be a sensitive index that fully captures the response signals of a community to removal treatments, particularly as changes in species evenness were not accounted for [26]. Another explanation is that the removal treatments often target the aboveground parts of plants, whereas the large roots of target species are more resilient and likely to survive the treatments. Consequently, although there might seem to be an available niche from an aboveground perspective, belowground dominance remains unaffected, resulting in no actual opening of niche opportunities that promotes species richness.

Similarly, the AR and SR treatments had distinct ecosystem impacts. Regarding the AR treatment, since the removal of *B. sinocompressus* had a greater impact on biomass production compared to the removal of *P. secundiflora*, our finding corroborates the mass ratio hypothesis that the dominant species with high abundance relative to other species has proportionate community and ecosystem impacts [27,28]. Regarding the SR treatment, although the sparse species, *P. secundiflora*, had a higher individual biomass than that of *B. sinocompressus*, its overall contribution to total biomass production was smaller due to its substantially lower abundance. Regardless of which species was removed, however, removal treatments still had significant community impacts after controlling for the effect of the removed biomass. Our results also showed that complete biomass compensation did not take place, and DR treatment had the lowest compensation response. Similar findings were observed in the studies conducted by Pan et al. [29] and Li et al. [30], where the removal of the dominant species resulted in a notable reduction in biomass. Furthermore, our results are consistent with those of Elumeeva et al. [31], illustrating that in high-altitude grasslands, full biomass compensation was not attained following the removal of the dominant species. One explanation is that short-term responses mostly involve partial compensation, as opposed to long-term experiments where full compensation may be achieved [32]. Meanwhile, the cold temperature at high altitudes may substantially constrain the effectiveness of community compensation, and high-altitude plant communities are likely to be functionally vulnerable [33,34]. Furthermore, our results suggest that soil fertility did not undergo significant alterations after the removal of the target species with comparable biomass contributions. One possible explanation is that soil water movement may have played a significant role in facilitating the transfer of soil nutrients, particularly through the diffusion and redistribution of nutrients following rainfall events. Additionally, some studies indicate that fertilization had no discernible impact on biomass production in high-altitude grasslands [35]. Given that the N and P content in the soil of our experimental sites falls within the medium to high range (N: 4.66–12.62 g/kg, P: 0.62–0.65 g/kg) when compared to N and P values reported by other studies [19], so the high-altitude grasslands in our study site may not necessarily experience nutrient limitations. Chaves and Smith [36] similarly confirmed through removal experiments that resource availability was not a limiting factor in the compensatory effects observed within plant communities following the loss of the dominant species. In addition, short-term responses mostly involve partial compensation, as opposed to long-term experiments where full compensation may be achieved [21].

We acknowledge that our experimental duration is relatively brief, which potentially limits the attainment of more substantial experimental findings. However, given that the alpine vegetation in our experimental site is accessible to both wild and domestic herbivores, an extended experiment duration could pose challenges in distinguishing the specific impact of the dominant plant species from that of herbivores on the alpine vegetation. Furthermore, even studies conducted over extended periods often yield varying findings, and full compensation is not consistently attained [31,36,37]. This could be attributed to variations in nutrient uptake efficiency among different species in high-altitude grasslands [38]. It might also be due to the absence of species in the experimental community with identical or comparable functions that could substitute for the role of the original dominant species. Therefore, while some exhibited substantial biomass compensation, the probability of achieving complete compensation remains rather low.

## 4. Materials and Methods

### 4.1. Study Area

The study site is located in the Shangri-La region of Yunnan Province, China, known for its typical alpine grassland ecosystem (Figure 5). It has an average elevation of 3450 m, and it experiences a cool temperate climate characterized by abundant rainfall in summer and autumn, and dry conditions in winter. Based on the meteorological data provided by the China Meteorological Administration, the annual average temperature is approximately 5.5 °C, with the warmest month having an average temperature of 13.2 °C and the coldest month having an average temperature of −3.8 °C. The region experiences considerable diurnal temperature variations, intense solar radiation, and an annual precipitation of approximately 632 mm [39,40]. The study site is situated near Pudacuo National Park (27°48′ N, 99°59′ E, 3560 m a.s.l), covered by typical alpine plant communities. Specifically, *B. sinocompressus* is an abundant species with high abundance, and *P. secundiflora* is a sparse species with small population sizes and large individual biomass. Other species that are common in this site include *Prunella vulgaris* L., *Ranunculus japonicus* Thunb. and *Sanguisorba filiformis* Hand.-Mazz. There are also sparse species like *P. secundiflora* but with lower individual biomass. These species are mostly perennial vegetation and exhibit relatively high growth rates during the summer months.

### 4.2. Experimental Design

We conducted a removal experiment in a randomly selected grassland site with flat terrain and similar species composition early in the growing season. We established a total of six replicate sets of 1 m × 1 m plots [41], and each plot separated by a minimum of 1 m. Within each plot a total of four 0.5 m × 0.5 m subplots [36,41] were divided (a total of 24 subplots) and randomly subjected to four treatments as follows: (1) The removal of the abundant species, *B. sinocompressus* (AR); (2) the removal of the sparse species, *P. secundiflora* (SR); (3) the double removal of *B. sinocompressus* and *P. secundiflora* (DR); and (4) a control group with no removal treatment (NR). This design maximizes the likelihood that the experimental area will have a similar species composition. To prevent soil surface disturbance that could affect the growth of other species, we used scissors to remove the aboveground portions of the target species.

### 4.3. Biodiversity and Soil Chemical Characterization

We conducted a vegetation survey several months after the removal. During this follow-up, we documented species richness and the abundance of each species within each plot. Moreover, we visually estimated the coverage of each species. It is worth noting that due to spatial heterogeneity and variations in species composition and abundance across the different communities, some species were either absent or had very low numbers. Consequently, these species were included for diversity estimation but not for biomass measurement. After completing species counts and coverage estimation, we collected samples of plant species from each plot, focusing solely on their aboveground portions. To avoid interactions between the treatment groups, we chose to sample treatments within the corners of each treatment group near the peripheral side. The collected samples were sealed in clean plastic bags for further processing. We air-dried the plant samples and placed them in an oven at 90 °C for 10 h before recording their weights [20]. Following the collection and sealing of the plant samples, we obtained soil samples corresponding to the different removal treatments. These soil samples from each subplot were placed in sealed bags and transported to the laboratory. Soil samples were collected from 10 cm below ground level and 200 g from each subplot group [42], for a total of 24 soil samples. In the lab, we used an automated nitrogen analyzer (Kjeldahl Nitrogen Determinator, Model K9840) to measure the total nitrogen (N) content in the soil and employed the sodium hydroxide fusion–molybdenum antimony colorimetric method (UV–visible spectrophotometer, model TU-1900) to determine the total phosphorus (P) content in the soil.

### 4.4. Statistical Analysis

To assess the impacts of the removed species on the alpine community structure, we measured vegetation coverage, species richness, Pielou’s evenness, and the Shannon diversity index. The Vegan package was used to calculate diversity metrics. To assess the impacts of the removed species on the ecosystem functions of the alpine grasslands, we measured the plants’ aboveground biomass as a proxy for productivity. Also, soil N and P content were analyzed. Furthermore, to test Grime’s hypothesis that a greater removal of biomass would lead to a more significant ecological impact, and to determine the community and ecosystem impacts of removal treatments after controlling for the effect of the removed biomass, an analysis of covariance was conducted, with removal treatments as the explanatory variables, the removed biomass as the covariate, and the properties of the community structure and ecosystem functions as the response variables.

To account for multiple comparisons and control the family-wise error rate, we applied the Bonferroni correction. Although our sample size is relatively small, the number of comparisons made warranted a conservative approach to reduce the risk of Type I errors. The Bonferroni correction was chosen to ensure the robustness and reliability of statistical inferences. Normality of residuals, and homogeneity of variances were checked to ensure that they met the assumptions of statistical analysis using the broom package. Post hoc analysis was performed with a Bonferroni adjustment to identify which treatment groups were different using the emmeans package. All statistical analyses were conducted in the R statistics 4.2.0 platform (R Core Development Team 2021).

## 5. Conclusions

Our findings support the mass ratio hypothesis that ecosystem functions such as biomass production depend on the highly numerous dominant species within a community, even if its individual biomass is lower than certain sparse species. Also, the dominant species exerts a strong community impact due to its spatial dominance. Overall, dominant plants play pivotal roles in alpine grasslands, and their losses can alter species interactions, change ecological processes, and disrupt ecosystem functions. Therefore, it is crucial to enhance our capacity to evaluate their vulnerability to environmental changes, and implement measures to mitigate the adverse effects on the provisioning services offered by the alpine grasslands.

## Figures and Tables

**Figure 1 plants-13-01756-f001:**
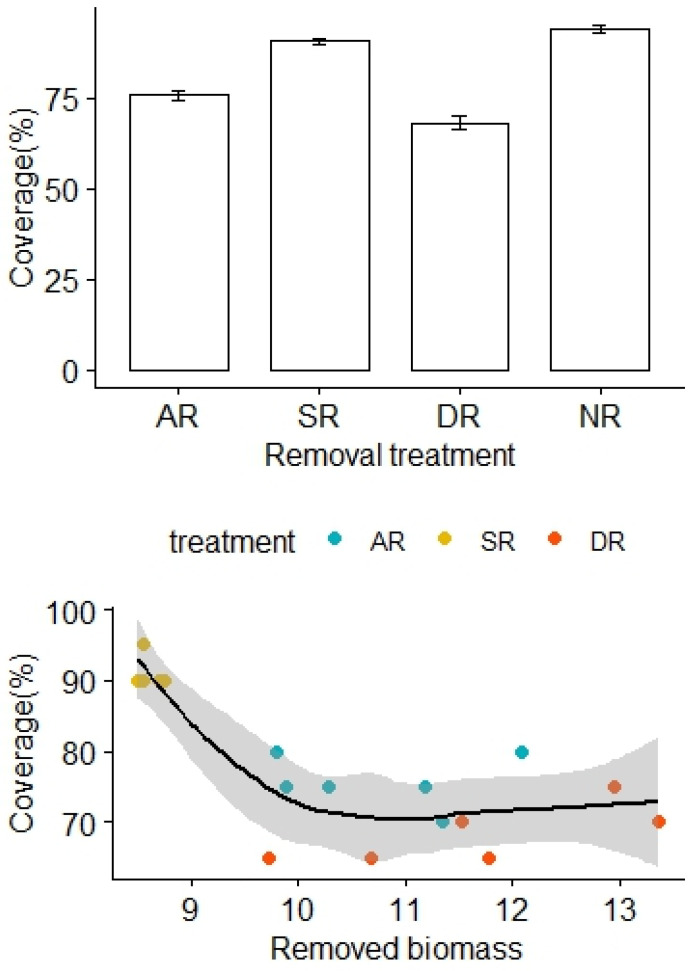
Effects of the removal treatments and removed biomass (in grams) on vegetation coverage of the alpine grasslands. AR: the removal of abundant species; SR: the removal of sparse species; DR: the double removal of AS and SS; NR: the control treatment without plant removals. Values are mean ± 1 standard error.

**Figure 2 plants-13-01756-f002:**
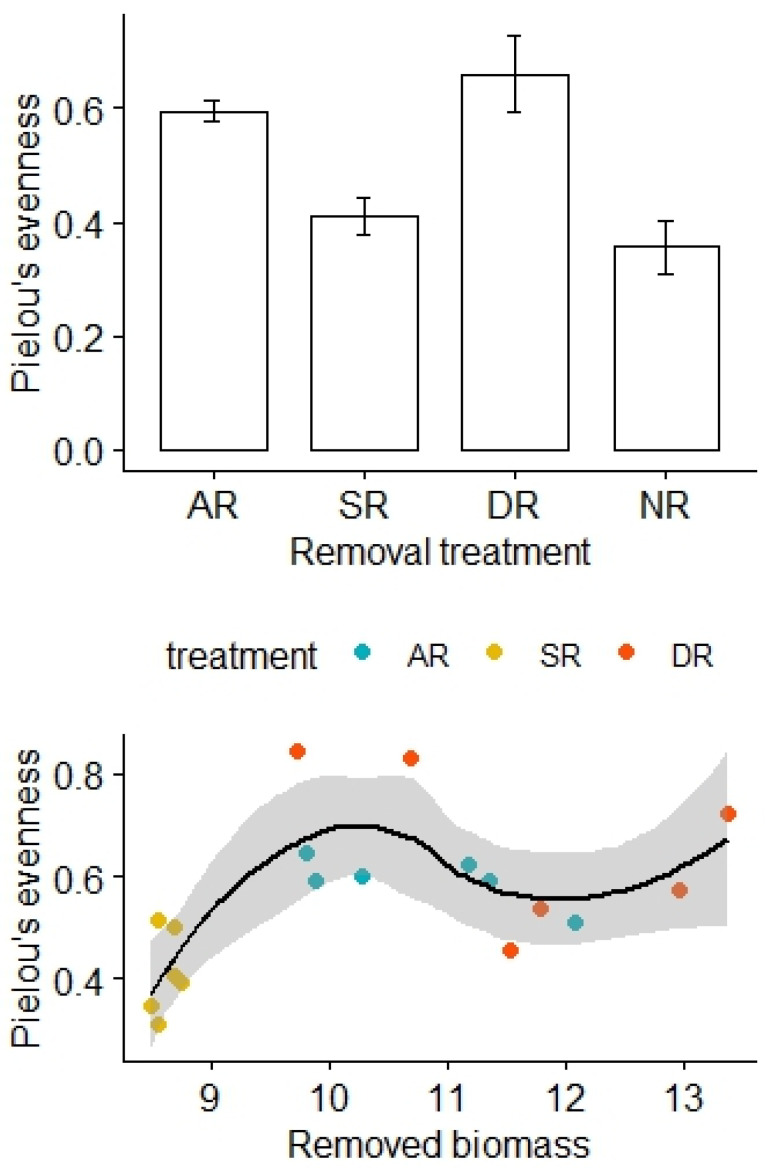
Effects of the removal treatments and removed biomass (in grams) on Pielou’s evenness of the alpine grasslands. AR: the removal of abundant species; SR: the removal of sparse species; DR: the double removal of AS and SS; NR: the control treatment without plant removals. Values are mean ± 1 standard error.

**Figure 3 plants-13-01756-f003:**
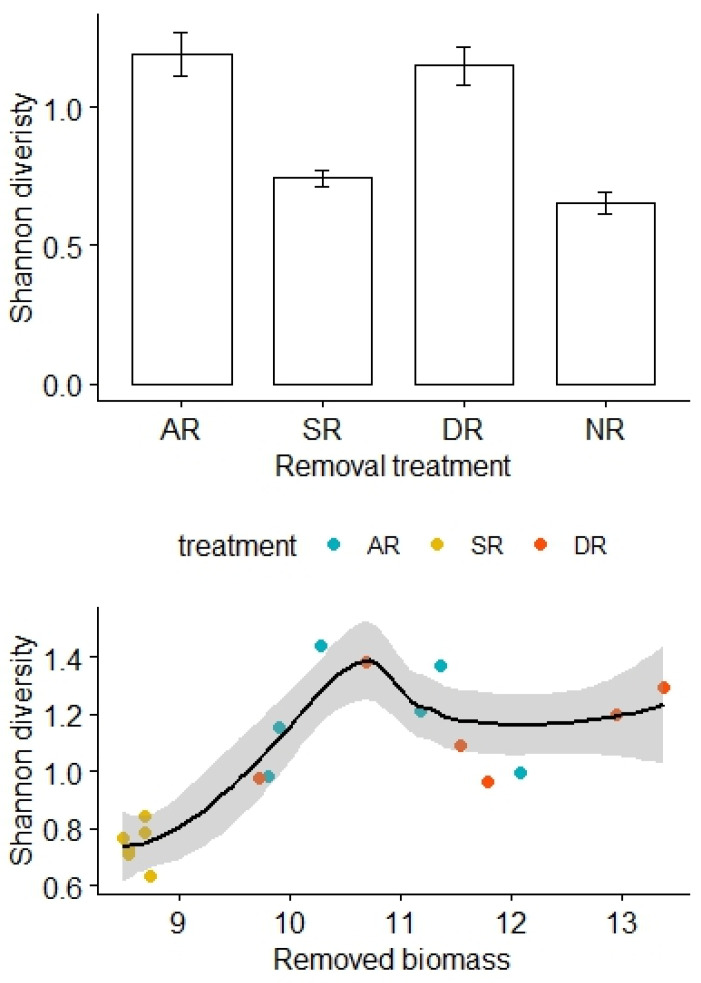
Effects of the removal treatments and removed biomass (in grams) on the Shannon diversity of alpine grasslands. AR: the removal of abundant species; SR: the removal of sparse species; DR: the double removal of AS and SS; NR: the control treatment without plant removals. Values are mean ± 1 standard error.

**Figure 4 plants-13-01756-f004:**
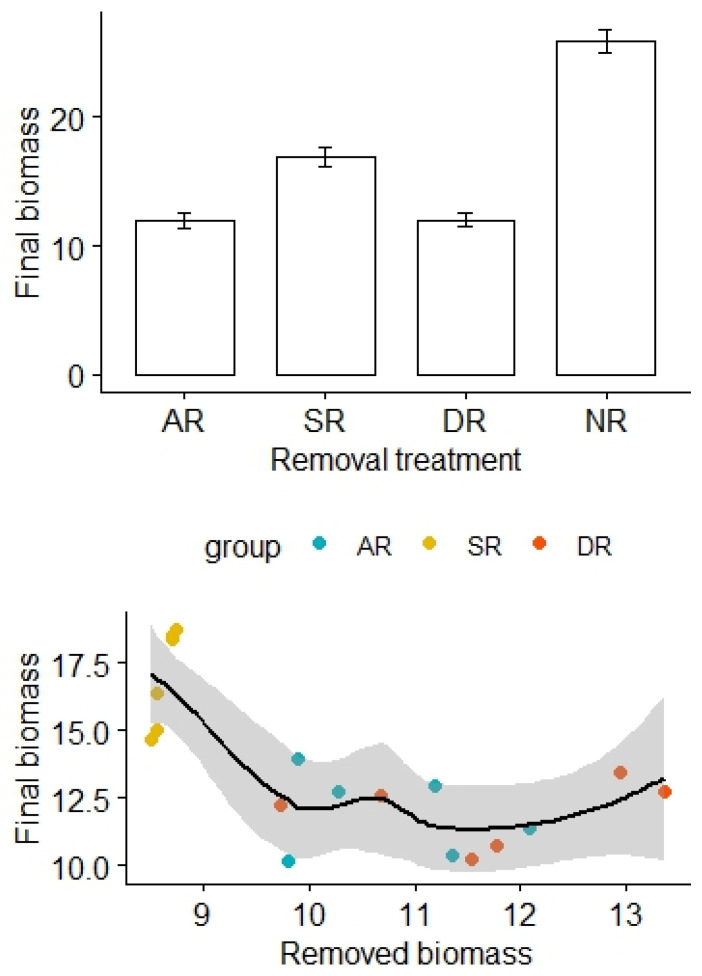
Effects of the removal treatments and removed biomass (in grams) on aboveground biomass of the alpine grasslands. AR: the removal of abundant species; SR: the removal of sparse species; DR: the double removal of AS and SS; NR: the control treatment without plant removals. Values are mean ± 1 standard error.

**Figure 5 plants-13-01756-f005:**
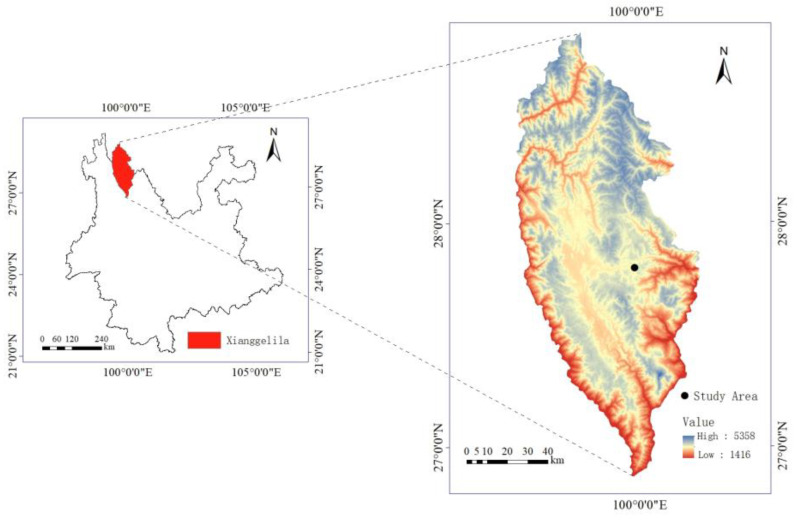
Information on the geographic location of the study area.

**Table 1 plants-13-01756-t001:** Analysis of covariance (ANCOVA) summary table for vegetation coverage by removed biomass and treatment settings. (SS: sum of squares, MS: mean square, * *p* < 0.05).

	SS	df	MS	F	*p*
Removed biomass	895.5	1	895.5	84.84	<0.01 *
Treatment	706.1	2	353	33.45	<0.01 *
Removed biomass/Treatment	21.7	2	10.9	1.03	0.387
Residuals	126.7	12	10.6		

**Table 2 plants-13-01756-t002:** Analysis of covariance (ANCOVA) summary table for Pielou’s evenness by removed biomass and treatment settings. (SS: sum of squares, MS: mean square, * *p* < 0.05).

	SS	df	MS	F	*p*
Removed biomass	0.072	1	0.072	6.301	0.027 *
Treatment	0.159	2	0.079	6.90	0.01 *
Removed biomass/Treatment	0.005	2	0.003	0.215	0.809
Residuals	0.138	12	0.012		

**Table 3 plants-13-01756-t003:** Analysis of covariance (ANCOVA) summary table for the Shannon diversity by removed biomass and treatment settings. (SS: sum of squares, MS: mean square, * *p* < 0.05).

	SS	df	MS	F	*p*
Removed biomass	0.509	1	0.509	18.327	<0.001 *
Treatment	0.233	2	0.117	4.195	0.031 *
Removed biomass/Treatment	0.008	2	0.004	0.137	0.873
Residuals	0.333	12	0.028		

**Table 4 plants-13-01756-t004:** Analysis of covariance (ANCOVA) summary table for final biomass by removed biomass and treatment settings. (SS: sum of squares, MS: mean square, * *p* < 0.05).

	SS	df	MS	F	*p*
Removed biomass	58.83	1	58.83	36.25	<0.01 *
Treatment	40.74	2	20.37	12.55	<0.01 *
Removed biomass/Treatment	16.85	2	8.42	5.19	0.02 *
Residuals	19.48	12	1.62		

**Table 5 plants-13-01756-t005:** Analysis of covariance (ANCOVA) summary table for soil nitrogen by removed biomass and treatment settings. (SS: sum of squares, MS: mean square).

	SS	df	MS	F	*p*
Removed biomass	18.58	1	18.58	0.50	0.49
Treatment	78.5	2	39.27	1.06	0.38
Removed biomass/Treatment	14.6	2	7.31	0.20	0.82
Residuals	444.8	12	37.07		

**Table 6 plants-13-01756-t006:** Analysis of covariance (ANCOVA) summary table for soil phosphorous by removed biomass and treatment settings. (SS: sum of squares, MS: mean square).

	SS	df	MS	F	*p*
Removed biomass	0.0061	1	0.0061	0.04	0.85
Treatment	0.5447	2	0.2724	1.74	0.22
Removed biomass/Treatment	0.2391	2	0.1195	0.76	0.49
Residuals	1.8825	12	0.1569		

## Data Availability

The raw data supporting the conclusions of this article will be made available by the authors on request.

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
