# Peer review of "Short-Term Responses of Alpine Vegetation to the Removal of Dominant versus Sparse Species"

_plants, 2024, doi:10.3390/plants13131756_

Round 1

Reviewer 1 Report

Comments and Suggestions for Authors

This article addresses a problem of theoretical and practical importance for nature conservation. Without going into the merits of the article, I would like to point out the shortcomings and, in my opinion, the points to be corrected. 

1. No author affiliations are listed. If the authors do not represent an institution (independent researchers), this should be stated or written as "unaffiliated". 

2. When referring for the first time to the scientific names of the species studied and other plant species, I recommend that the authors of the taxa are also mentioned, so that there is no ambiguity about their taxonomic identity. 

3. In my opinion, the last paragraph (lines 59-81) of the Introduction is not well conceived and, on certain extant, is not part of the Introduction. It contains information from the results (on biomass, density of individuals), methods (on experimental design) and discussion. I think that this part of the Introduction should refer to the literature, and may formulate a hypothesis or a research aim and present the questions to be addressed. In the current version of the manuscript, the questions are formulated, but are too intertwined with the whole text and, in my opinion, the article addresses more questions than stated. 

4. Throughout the text of the results, there should be spaces between the equal sign and the symbols, as there are with the < sign. 

5. In statistics, the Bonferoni correction is not usually referred to as an 'adjustment' (lines 103, 110 etc.). It is questionable whether the Bonferoni correction was really necessary in this case, as the sample is small. The Bonferroni correction is applied when there are large samples of data and the aim is to avoid the resulting pseudo-differences. I think that the authors should justify the use of this correction in the methods section. Furthermore, as the methods are at the end of the article, the results should clearly state that the Bonferoni correction has been applied (in the indicated and other lines). 

6. I completely disagree with the inclusion of tables and figures in a separate subsection (2.1). They must be integrated into the main text of the Results section.

7. In the figures, the graphs must be lettered and it must be clearly indicated what each element means. Do the whiskers refer to the standard error or standard deviation? Also avoid abbreviations in figure captions, as it is very confusing to look up where they are written (especially when the methods are in an unusual place).

8. How to understand the "removed biomass" on the x-axis of the curves (Figures 1-4)? What do the numbers mean? Is it the amount of biomass? If so, it must be clearly stated so that the reader does not have to guess about the authors' intentions.

9. The full name of the species studied should be mentioned at least once in the discussion, not just the abbreviations (lines 159-160).

10. As there were four experimental treatments per square metre (0.25 m2 each), do the authors consider that the removal of plants and biomass could have influenced the results of the control and other variants of the experiment? This question should be addressed in the discussion.

Comments on the Quality of English Language

Minor revisions of language is required

Author Response

Dear Reviewer 1:

Thank you very much for the review of our manuscript. Your valuable comments and suggestions have helped us improve the quality of the manuscript. We have revised the manuscript accordingly, and our responses to the comment are described below.

  1. No author affiliations are listed. If the authors do not represent an institution (independent researchers), this should be stated or written as "unaffiliated". 

Response: We apologize for this oversight. The authors’ affiliations have now been added to the manuscript.

  1. When referring for the first time to the scientific names of the species studied and other plant species, I recommend that the authors of the taxa are also mentioned, so that there is no ambiguity about their taxonomic identity. 

Response: Thank you for your insightful comments. We have revised the manuscript to include the names of the taxonomists who originally described the species when mentioning their scientific names for the first time. Please see L75-76, L162 and L236-237.

  1. In my opinion, the last paragraph (lines 59-81) of the Introduction is not well conceived and, on certain extant, is not part of the Introduction. It contains information from the results (on biomass, density of individuals), methods (on experimental design) and discussion. I think that this part of the Introduction should refer to the literature, and may formulate a hypothesis or a research aim and present the questions to be addressed. In the current version of the manuscript, the questions are formulated, but are too intertwined with the whole text and, in my opinion, the article addresses more questions than stated. 

Response: Thank you for your constructive comments. We have revised the relevant paragraph in the introduction to better align with its intended purpose. Please see L60-83.

  1. Throughout the text of the results, there should be spaces between the equal sign and the symbols, as there are with the < sign. 

Response: Thank you very much for bringing this to our attention. We have carefully reviewed the manuscript, and adjusted the text in the results section to incorporate spaces around the equal sign and other mathematical symbols to ensure consistency and readability.

  1. In statistics, the Bonferoni correction is not usually referred to as an 'adjustment' (lines 103, 110 etc.). It is questionable whether the Bonferoni correction was really necessary in this case, as the sample is small. The Bonferroni correction is applied when there are large samples of data and the aim is to avoid the resulting pseudo-differences. I think that the authors should justify the use of this correction in the methods section. Furthermore, as the methods are at the end of the article, the results should clearly state that the Bonferoni correction has been applied (in the indicated and other lines). 

Response: Thank you for your valuable comments. We have revised the manuscript to address your concerns regarding the use of the Bonferroni correction. Please see L287-291.

  1. I completely disagree with the inclusion of tables and figures in a separate subsection (2.1). They must be integrated into the main text of the Results section.

Response: Thank you for your helpful suggestions. We have revised the manuscript to integrate the tables and figures directly into the text of the results section, rather than presenting them in a separate subsection.

  1. In the figures, the graphs must be lettered and it must be clearly indicated what each element means. Do the whiskers refer to the standard error or standard deviation? Also avoid abbreviations in figure captions, as it is very confusing to look up where they are written (especially when the methods are in an unusual place).

Response: Thank you for your valuable comments on the graphical part of our paper. We have labeled the whiskers in the manuscript: values are mean ± 1 standard error. We have opted to use abbreviations in the figures due to the lengthy nature of the full names, which would otherwise occupy a significant amount of space, with the full names provided in Figure legends.

  1. How to understand the "removed biomass" on the x-axis of the curves (Figures 1-4)? What do the numbers mean? Is it the amount of biomass? If so, it must be clearly stated so that the reader does not have to guess about the authors' intentions.

Response: Thank you for pointing out the need for clarification regarding the "removed biomass" on the x-axis of Figures 1-4. Here, the numbers indicate the removed biomass in grams. We have updated the manuscript to explicitly state the meaning of these numbers. Please refer to Figures 1-4 for the revised information..

  1. The full name of the species studied should be mentioned at least once in the discussion, not just the abbreviations (lines 159-160).

Response: Thanks for the helpful suggestion. We have revised the discussion section to include the full scientific names of the studied species at least once, instead of using only abbreviations. Please see L161 and L168.

  1. As there were four experimental treatments per square metre (0.25 m2 each), do the authors consider that the removal of plants and biomass could have influenced the results of the control and other variants of the experiment? This question should be addressed in the discussion.

Response: Thanks for the comments. To prevent the impact of performing removal treatments on other plots, we sampled a corner area close to the periphery within each plot as described in the manuscript. Please see L261-263.

Reviewer 2 Report

Comments and Suggestions for Authors

Wang and Li's study focuses on the particular influence of dominant species in alpine grasslands, pointing out that their effect on the community and ecosystem corresponds to their abundance.

Introduction: needs to be resized. Add at least one paragraph to introduce the reader to the topic. Then present the proposed goals more succinctly.

Results: should alternate explanatory text with figures or tables. Redo this section.

 Discussions are well presented to the standard required by the journal. There is not much to comment on. 

line 170 AR and SR treatments should be discussed separately.

I also highlight the limitations of the study and future studies which are appreciated.

Materials and Methods: It is necessary to present a map of the studied area.

219-239 references are required.

 Conclusions are brief, highlighting the achievement of the proposed aims.

We noted the low number of references and I am not very happy with this. Usually, a valid study is based on much more literature.

Author Response

Dear Reviewer 2:

Thank you very much for the review of our manuscript. Your valuable comments and suggestions have helped us to improve the quality of the manuscripts. We have revised the manuscript accordingly, and our responses to the comment are described below.

Introduction: needs to be resized. Add at least one paragraph to introduce the reader to the topic. Then present the proposed goals more succinctly.

Response: Thank you very much for reviewing our paper and for your valuable suggestions. We have added one paragraph to highlight a problem with both theoretical and practical importance for the conservation of alpine grasslands, which also allows us to better present the proposed goals.  Please see L60-83.

Results: should alternate explanatory text with figures or tables. Redo this section.

Response: Thank you for the helpful suggestion. We have revised the manuscript to integrate the tables and figures directly into the text of the results section to improve the clarity and readability of the results presentation.

line 170 AR and SR treatments should be discussed separately.

Response: Thank you for pointing this out. Based on your comments, we have discussed the impact of AR and SR treatments separately. Please see L178-187.

219-239 references are required.

Response: Thank you for reviewing our paper and for your valuable suggestions. We have added relevant references. Please see L218-219,L236 and L239.

We noted the low number of references and I am not very happy with this. Usually, a valid study is based on much more literature.

Response: Thanks for the constructive comments. We have added more references to ensure that our study is grounded in existing research findings.

Round 2

Reviewer 1 Report

Comments and Suggestions for Authors

The authors have made significant revisions to the manuscript and have taken into account most of the comments made in the review. Only a few minor shortcomings remain:

1. The authors of the manuscript added the standard forms of the names of the authors who described the taxa, but made a lot of mistakes. This suggests that the authors are not familiar with the rules of nomenclature. In botanical nomenclature, authors of taxa cannot be cited differently than in the nomenclature papers, and the authors of some taxa are confused.  

a) Blysmus sinocompressus must be Blysmus sinocompressus Tang & F.T.Wang

b) Primula secundiflora (Franch.) must be Primula secundiflora Franch. (without parentheses; adding parentheses you change authorship! See International Code for Nomenclature of algae, fungi and plants)

c) Ranunculus japonicus (Thunb.) must be Ranunculus japonicus Thunb., etc. Please check all plant names and their authors.

2. Technical and spelling errors still remain in the text and must be corrected.

Comments on the Quality of English Language

Minor corrections.

Author Response

Dear Reviewer 1:

Thank you very much for your thorough and insightful comments on our manuscript, which have been incredibly helpful in improving the clarity and quality of our work. We appreciate the time and effort you invested in reviewing our paper and providing such detailed and constructive suggestions.

  1. The authors of the manuscript added the standard forms of the names of the authors who described the taxa, but made a lot of mistakes. This suggests that the authors are not familiar with the rules of nomenclature. In botanical nomenclature, authors of taxa cannot be cited differently than in the nomenclature papers, and the authors of some taxa are confused.  
  2. a) Blysmus sinocompressus must be Blysmus sinocompressus Tang & F.T.Wang
  3. b) Primula secundiflora (Franch.) must be Primula secundiflora Franch. (without parentheses; adding parentheses you change authorship! See International Code for Nomenclature of algae, fungi and plants)
  4. c) Ranunculus japonicus (Thunb.) must be Ranunculus japonicus Thunb., etc. Please check all plant names and their authors.

Response: Thank you for the detailed and constructive feedback. We apologize for the errors in the taxon author citations. We have carefully revised the manuscript. Please see L70-71, L159, and L235-236.

  1. Technical and spelling errors still remain in the text and must be corrected.

Response: Thank you for your feedback. We have thoroughly proofread the manuscript, identifying and correcting technical and spelling errors to ensure clarity and accuracy.

Reviewer 2 Report

Comments and Suggestions for Authors

The authors have taken into account the reviewer's comments and the manuscript has been improved. 

Material and Methods: A map of the study area should be added.

I have no further comments.

Author Response

Dear Reviewer 2:

Thank you very much for your thorough and insightful comments on our manuscript, which have been incredibly helpful in improving the clarity and quality of our work. We appreciate the time and effort you invested in reviewing our paper and providing such detailed and constructive suggestions.

Material and Methods: A map of the study area should be added.

Response: Thank you for your valuable suggestions. We have added a map of the study area in the "Materials and Methods" section.  Please see L239.
